# Air Pollution and Incidence of Lung Cancer by Histological Type in Korean Adults: A Korean National Health Insurance Service Health Examinee Cohort Study

**DOI:** 10.3390/ijerph17030915

**Published:** 2020-02-02

**Authors:** Da Hye Moon, Sung Ok Kwon, Sun-Young Kim, Woo Jin Kim

**Affiliations:** 1Department of Internal Medicine, Kangwon National University, Chuncheon 24341, Korea; ansekgo@naver.com; 2Biomedical Research Institute, Kangwon National University Hospital, Chuncheon 24289, Korea; kamelon@hanmail.net; 3Department of Cancer Control and Population Health, Graduate School of Cancer Science and Policy, National Cancer Center, Goyang-si 10408, Korea; sykim@ncc.re.kr

**Keywords:** lung cancer, air pollution, adenocarcinoma

## Abstract

Studies have reported associations between long-term exposure to ambient air pollution and lung cancer. However, there have been inconsistent reports of such associations with lung cancer by histological type. Thus, the aim of this study was to assess the association of long-term exposure to particulate matter with an aerodynamic diameter up to 10 μm (PM_10_) and nitrogen dioxide (NO_2_) with lung cancer incidence by histological subtype in South Korea. This population-based cohort study included 6,567,909 cancer-free subjects from the Korean National Health Insurance Service (NHIS) database for 2006–2007. We linked population data to Korea Central Cancer Registry data to confirm lung cancer incidence for 2006–2013. Individual exposures to PM_10_ and NO_2_ were assessed as five-year average concentrations predicted at subjects’ district-specific home addresses for 2002–2007. We divided these exposures into two categories based on the 75th percentile. Cox proportional hazards models were used to estimate hazard ratios (HRs) of lung cancer incidence for the upper 25% exposure compared to the low 75% by histological subtypes at diagnosis after adjusting for potential confounders. A total of 27,518 lung cancer were found between 2006 to 2013. The incidence of lung cancer was higher in males, smokers, drinkers and subjects with chronic obstructive pulmonary disease. Overall, we did not find an increased risk of lung cancer with higher exposure to PM_10_ or NO_2_. However, high exposure to PM_10_ was associated with increased risk of adenocarcinoma in comparison with lower exposure in males and current smokers (HR, 1.14; 95% CI, 1.03–1.25). This study showed that long-term air pollution exposures were associated with an elevated risk of lung adenocarcinoma in male smokers in Korea.

## 1. Introduction

Lung cancer is one of the most common cancers with poor prognosis and high mortality. Smoking is the main cause of lung cancer. Occupational exposure and residential radon are established as common environmental risk factors. In addition, several studies have reported associations between long-term exposure to ambient air pollution and lung cancer. Seventeen cohort studies in nine European countries showed an association between residential exposure to particulate matter air pollution and the risk for lung cancer. One study using a random 5% sample of China Urban Employee Basic Medical Insurance (UEBMI) also showed the increased risk of lung cancer associated with increasing long-term PM2.5 air pollution exposure from 1 January 2010 to 31 December 2016 [1,2]. Several prospective cohort studies in North America and Europe have evaluated the effects of long-term exposure to outdoor air pollution and demonstrated associations between ambient air pollution exposure with respiratory disease, cardiovascular disease and lung cancer [3,4].

A recent Korean lung cancer epidemiology study showed that the most frequent histological subtype was adenocarcinoma [5]. Although squamous cell carcinoma was the most common subtype in male patients, adenocarcinoma was the most common subtype in female patients [6]. A few studies have looked at variations in lung cancer histology in relation to fine particulate matter, and only one study has found no clear relationships between exposure to fine particulate matter and specific histologic cancer type [7,8,9].

The aim of this study was to assess the association of long-term exposure to particulate matter with aerodynamic diameter up to 10 μm (PM_10_) and nitrogen dioxide (NO_2_) with lung cancer incidence by histological subtype in South Korea, using data link between health insurance service data and cancer registry data. 

## 2. Materials and Methods

### 2.1. Study Design and Participants 

We collected data from the medical insurance claims and biennial health examination results of a standardized cohort sampled from the Korean National Health Insurance Service (NHIS) database. The NHIS, a single-insurer system in 2000, was launched by integrating more than 366 medical insurance organizations, for efficient system operation in Korea. The NHIS provides benefits for prevention, diagnosis, disease and injury treatment, as well as rehabilitation, births, deaths and health promotion. The NHIS database contained the personal information, demographics and medical treatment data for Korean citizens who were categorized as insured employees, insured self-employed individuals or medical aid beneficiaries [10]. From NHIS records, we obtained the following information about individuals aged 30 years or more who went to the Korean national health examination service between 1 January 2006 and 31 December 2007 (*n* = 12,907,369). We excluded 6,339,381 individuals who had been diagnosed with any cancer, indicated by the 10th revision of International Classification of disease (ICD-10) C code or D code received in the previous five years for 2002-2006 and those under the age of 30. Based on cancer statistics studies in Korea, the incidence rate of lung cancer has increased from those in their 30s or older, so we excluded those in participants [11,12,13]. 

Among the selected 6,567,988 subjects, 79 diagnosed with lung cancer in connection with the cancer registration database of the Korea Central Cancer Registry (KCCR) from 2003 to 2007 were excluded. The Korean National Cancer Center Community Cohort was begun as a part of the Korean Multi-center Cancer Cohort in 1993 and has been continuously developed and funded by the National Cancer Center since 2001. These databases provide the Korean Central Cancer Registry and cause of death, which are reliable registries covering the entire population and potential risk factors for cancer, including environmental factors, genetic factors and other chronic diseases [14]. Thus, a total of 6,567,909 participants were included for the final analysis as shown in Figure 1. 

This population-based cohort study included 6,567,909 people who were cancer-free in the 2006–2007 Korean National Health Insurance Service (NHIS) database. We matched ICD codes (C34) to KCCR data to confirm lung cancer incidence from 2006 to 2013.

### 2.2. Exposure Assessment 

Individual-level exposure to PM_10_ or NO_2_ was assessed based on predicted district-specific annual average concentrations over South Korea entirely obtained from a previously developed air pollution prediction model [15]. This geo-statistical model was developed by using air quality monitoring network data of the National Institute of Environmental Research (NIER) and various geographic predictors computed in Geographic Information System. Individual exposures were computed as five-year averages using predicted annual average PM_10_ and NO_2_ concentrations from 2002 to 2007 in subject’s residential districts. Subjects were divided into two exposure groups based on the 75th percentile. 

### 2.3. Statistical Analysis

Cox proportional hazards models were used to estimate the hazard ratio (HR) of lung cancer incidence in total and by histologic type at diagnosis for PM_10_ and NO_2_. Individual variables included gender, age, body mass index (BMI), smoking and drinking habits, physical activity, history of chronic obstructive pulmonary disease (COPD), lung diseases due to external factors and interstitial lung disease. For some individual characteristics, we classified into the following categories: BMI (<18.5, 18.5–22.9, 23–24.9, 25–29.9, ≥30; Asian-Pacific cutoff points), smoking status (never, former, current-smoker), frequency of physical activity (non, 1–2/week, 3–4/week, 5–6/week, every day) and alcohol consumption (none, 2–3/month, 1–2/week, 3–4/week, every day).

Lung cancer was defined based on ICD-O-3 topography code C34 in the National Cancer Registry. Cancers defined morphologically as adenocarcinoma (8140/3), squamous cell carcinoma (8070/3), large cell carcinoma (8012/3, 8072/3, 8013/3, 8071/3, 8014/3, 8046/3) and small cell carcinoma (8041/3) and not specified (8250/3).

For risk analysis of lung cancer incidence, firstly, we used age and gender-adjusted models. Secondly, we adjusted for age, gender, income, BMI, smoking status, past lung diseases (COPD, lung diseases due to external factors and interstitial lung disease) and drinking.

All analyses were performed using the SAS statistical version 9.4 (SAS Institute, Cary, NC, USA). A *p*-value less than 0.05 was considered to indicate statistical significance.

### 2.4. Ethics Statement

The Institutional Review Board (IRB) of Kangwon National University Hospital approved the study protocol (IRB No. 2018-01-011) and waived the requirement for informed consent.

## 3. Results

### 3.1. Description of the Study Population and Lung Cancer Incidence

Table 1 shows the individual-level concentrations of PM_10_ and NO_2_ for the past five years (2002-2007). It was 55.8 mcg/m^3^ (Min to Max; 37.99 to 75.95), and 23.9 ppm (Min to Max; 4.16 to 42.66), for PM_10_ and NO_2_. The correlation between PM_10_ and NO_2_ concentration was 0.77. 

The general characteristics of the study subjects were presented in Table 2. The cohort, which was established as a target for the 2006-2007 health examination, was 6,567,909 persons. The cohort comprised 65.7% (4,314,480) of men and 34.3% (2,253,429) of women. About 80 percent of the study population was aged between 30 and 50: 28.3%, 30.6% and 21.1% for 30s, 40s and 50s, respectively. Non-smokers account for 59.4%. Past and present smokers accounted for 10.7% and 29.9%, respectively. The average age of current smokers was 17.1 (±10.9) pack/years. In the past, 1.3% (83,419) of chronic closed lung patients were diagnosed with respiratory diseases, 0.05% (3164) of lung diseases due to external factors and 7130 (1.0%) with interstitial lung disease were found.

Lung cancer incidence by histologic classification of study subjects is shown in Table 3. By 2006–2013, a total of 27,518 new cases of lung cancer were found. With the exception of small cell carcinoma in total lung cancer, the number of patients with non-small cell carcinoma was 24,031. By histologic classification, there were 7280 squamous cell carcinomas, 8223 adenocarcinomas, 314 large cell carcinomas and 3127 small cell carcinomas. Among other lung cancers, there were 1850 non-small cell carcinomas (Mcode, 8046/3) and 380 bronchioloalveolar carcinomas. By gender, 21,912 and 5606 incident lung cancers were documented among men and women, respectively.

By histologic classification, men were most likely to develop squamous cell carcinoma (6912), adenocarcinoma (5744) and small cell carcinoma (2836) while adenocarcinoma (2479) was the most common in women.

### 3.2. Factors of Demographic and Lifestyle Behavior Affecting the Development of Lung Cancer

The risk of total lung cancer according to the general information of the study subjects is presented in Table 4. Univariable analysis of individual factors affecting lung cancer incidence revealed that the higher incidence of developing lung cancer overall was higher in men than in women and in the lower BMI group than in the normal group. In smoking, the risk of lung cancer incidence in former smokers was higher compared to non-smokers (HR, 1.49; 95% CI, 1.43–155), with current smoking being the most important risk factor (HR, 2.04; 95% CI, 1.99–2.10). The risk of lung cancer was significantly higher in those with chronic closed lung disease or other lung diseases and in those with exercise or drinking daily. Women with a lower weight or overweight group had a higher risk of developing lung cancer than normal-weight people, while men in the overweight group had a lower risk of developing lung cancer. Analysis of factors affecting lung cancer incidence by histologic classification showed similar patterns to those for total lung cancer.

### 3.3. Air Pollution Exposure and Lung Cancer Incidence

Table 5 shows the risk of lung cancer incidence by systematic classification according to exposure to air pollution by smoking status. We did not observe an increased risk associated with PM_10_ or NO_2_ exposure. The 75th percentage level of air pollution was based on PM_10_ at 60.9 μg/m^3^ and NO_2_ at 32.1 ppm. In the final model, adjusting for other influencing factors, the highest exposure to PM_10_ in the top 25% percentile was significantly associated with increased risk of adenocarcinoma in comparison with lower exposure in the bottom 75% percentile (HR, 1.14; 95% CI, 1.03–1.25) in male smokers. A marginally significant association (*p* < 0.10) was found for non-small cell carcinoma (HR, 1.05; 95% CI, 0.99–1.11). According to gender, the male group had the same analysis result as all subjects. PM_10_ had no significant association with lung cancer incidence in women. However, higher estimated risks of other lung cancers (except for non-small cell carcinoma and adenocarcinoma, etc.) appeared to be marginally associated with NO_2_ exposure (HR, 1.11; 95% CI, 0.99–1.25) in women. 

## 4. Discussion

This study evaluated the association between air pollution and incidence of lung cancer by histological type in Korean adults. A Korean National Health Insurance Service Health Examinee Cohort study was established as a target for the 2006–2007 health examination participants (*n* = 6,567,909). By 2013, total lung cancer cases were 27,518, and non-small cell carcinoma cases were 24,031. We did not find that higher exposure to air pollution increased the risk of lung cancer overall. However, in sub-analysis, there was a significant association between high exposure to PM_10_ and increased risk of adenocarcinoma in male and current smokers (HR, 1.14; 95% CI, 1.03–1.25). 

The important risk factor of cigarette smoking in the etiology of lung cancer has been well established throughout numerous epidemiological studies [16,17,18]. This study also showed that the risk of total lung cancer in former smokers was 1.49 (95% CI = 1.43–155) compared to non-smokers and 2.04 (95% CI = 1.99–2.10) as a strong risk factor for current smokers. Some studies have shown that the histological type of lung cancer in Korea changed from the past to the present. A nationwide survey of lung cancer was first performed by the Korean Academy of Tuberculosis and Respiratory Diseases (KATRD) in 1998, in which histologically confirmed lung cancer patients (n = 3794) diagnosed in 1997 were retrospectively surveyed. Patient data were collected from 50 hospitals with more than 400 beds. Squamous cell carcinoma was the most frequent histologic type of lung cancer (44.7%), followed by adenocarcinoma (27.9%). Squamous cell carcinoma was attributed to a high rate of smoking and unfiltered cigarettes. The second nationwide survey of lung cancer was conducted by the Korean Association for Lung Cancer (KALC) in 2007, reporting 8788 patients diagnosed with lung cancer in 2005. Patient data were collected from 64 hospitals (more than 400 beds). The most common histological subtype was adenocarcinoma (48.7%), followed by squamous cell carcinoma (27.2%) and small cell lung cancer (11.5%). The leading histopathologic subtype had changed from squamous cell carcinoma to adenocarcinoma [19]. While squamous cell carcinoma was the leading subtype in males, adenocarcinoma showed the highest incidence in females [6]. 

Some studies have described that historically, the first wave of increase in adenocarcinoma is partially influenced by the introduction of novel diagnostic methods that can improve access to the lung periphery, where adenocarcinoma predominantly develops. The second wave of increase was related to changes in cigarette composition and design [20,21].

Several studies have proved the association between exposure at ambient air pollutants, including particulate matter with a diameter of fewer than 10 micrometers (PM10) and nitrogen dioxide (NO_2_), and the risk of lung cancer according to histological subtypes [7,22]. Our study showed that in male smokers, the highest exposure to PM_10_ was significantly associated with increased risk of adenocarcinoma in comparison with those having lower exposure (HR, 1.14; 95% CI, 1.03–1.25). The recently published findings are consistent with the results of the present study. The ESCAPE study found ORs for adenocarcinoma (based on 663 cases) and squamous cell carcinoma (322 cases), respectively, of 1.51 (95% CI: 1.10–2.08) and 0.84 (95% CI: 0.50–1.40) per 10 μg/m^3^ of PM_10_ [1]. The Nurses’ Health Study (NHS) cohort study in the USA examined the association between PM_10_ estimated with spatio-temporal models and lung cancer incidence among about 103,000 female nurses in the period 1994–2010. The HR for adenocarcinoma was 1.18 (95% CI: 0.97–1.45, 847 cases), which increased to 1.41 (95% CI: 0.95–2.09, 425 cases) when the analysis was restricted to never or former smokers who had quit at least 10 years previously [23]. The International Agency for Research on Cancer (IARC)’s systemic review and meta-analysis showed the lung cancer risk associated with exposure to PM in outdoor air, specifically PM_2.5_ and PM_10_. The meta-estimates for adenocarcinoma associated with PM_2.5_ and PM_10_ were 1.40 (95% CI: 1.07, 1.83) and 1.29 (95% CI: 1.02, 1.63), respectively [8]. This was similar to the result showing major shifts in the frequencies of different histological types of lung cancer, with substantial relative increases in adenocarcinoma and decreases in squamous cell carcinomas [24]. On the other hand, some studies have shown the association with other histological subtypes. According to the Environment and Genetics in Lung Cancer Etiology (EAGLE) study, a population-based case-control study performed in the period 2002-2005 in the north-west Italy, there was a steeper relationship for squamous cell carcinoma (fully adjusted OR2 1.44; 95% CI: 0.90–2.29) than for adenocarcinoma: (OR2 1.13; 95% CI: 0.79–1.72) [25]. In South Korea, one study was conducted as a case-control study, a matched case-control study with a 1:1 ratio, at three university hospitals in Seoul and Incheon, Korea. This study has found an increased risk of lung cancer incidence with residential exposure to ambient PM_10_ and NO_2_ in particular squamous cell carcinoma and small cell carcinoma in Korea. The association was slightly stronger for squamous cell carcinoma (OR = 1.15; 95% CI: 0.98–1.35, 188 cases) than for adenocarcinoma (OR = 1.09; 95% CI: 0.97–1.22, 559 cases) [26]. Although the studies are of the same nation, there are some differences. First of all, there must be a difference in study design. This study was a retrospective cohort study, On the other hand, Lamichhane et al. were analyzed as a case-control study. The second, Lamichhane et al. were conducted by extracting patients from three hospitals. However, this study was analyzed using big data from the Korean National Health Insurance Service (NHIS) and the Korea Central Cancer Registry (KCCR). It showed the relationship between the types of lung cancer and ambient air pollution in consideration of gender and smoking history. An American Cancer Society (ACS) study controlled for active and passive smoking levels has reported stronger associations between air pollution and lung cancer in males, whereas associations are more evident in former smokers [27]. 

In women, higher estimated risks of other lung cancers (except for non-small cell carcinoma and adenocarcinoma, etc.) appeared to be marginally associated with NO_2_ exposure (HR, 1.11; 95% CI, 0.99–1.25). Some studies have found that residence in proximity to heavy-traffic roads, or with exposure to NO_2_ (particularly when considering levels of exposure greater than 30 μg/m^3^), can increase the risk of lung cancer [28,29]. However, these studies did not show that the incidence of lung cancer to histological type or gender. 

One study showed that PM_10_ and NO_2_ exposures were associated with an increased risk of lung cancer. However, such risk did not differ between males and females [30,31]. However, in the current study, we did not observe any significant association in females.

Our study has several important limitations. First, in our study, exposure was assessed at the enrollment address. We could not evaluate relocation during the follow-up period. It might have led to misclassification of the exposure relevant to later development of lung cancer. 

Second, a large portion of cancers was classified into other cancer types (*n* = 8214). We extracted data from the Korea Central Cancer Registry from 2006 to 2013. This study design used retrospective data. Thus, some clinical information could not be extracted. In addition, the histological type of lung cancer could be uncertain. This analysis involved different parts of hospitals in South Korea. Although the cancer registry was based on extreme chart review, histological type could not be specialized. For an in-depth analysis of the study, precise registration of histological diagnosis with homogeneous methodology is needed. 

Third, many studies have shown a high relationship between occupational exposure and lung cancer incidence [32]. We collected the Korean National Health Insurance Service (NHIS) data about different variables, including information on participant’s identity and socioeconomic variables such as gender, residential area, type of health insurance, level of income, type and grade of disability registered, birth and death. However, such data could not reflect the occupational characteristics of the participants. 

Nevertheless, the strength of this study was that we could link health insurance data with cancer registry data. In addition, this study showed air pollution and lung cancer incidence corrected for other factors (age, gender, income, BMI, cigarette smoking history, history of lung disease, physical activity, alcohol consumption). This large-scale retrospective cohort study using Health Insurance and Cancer Registry data demonstrated associations between ambient air pollution and elevated risk of lung cancer according to its histological type, after controlling for confounding factors.

## 5. Conclusions

In conclusion, the results of this study showed that long-term air pollution exposures were associated with an elevated risk of lung adenocarcinoma in male smokers in Korea. Further studies are needed to determine the causes of differences in types of lung cancer according to gender and ambient air pollution in greater depth. 

## Figures and Tables

**Figure 1 ijerph-17-00915-f001:**
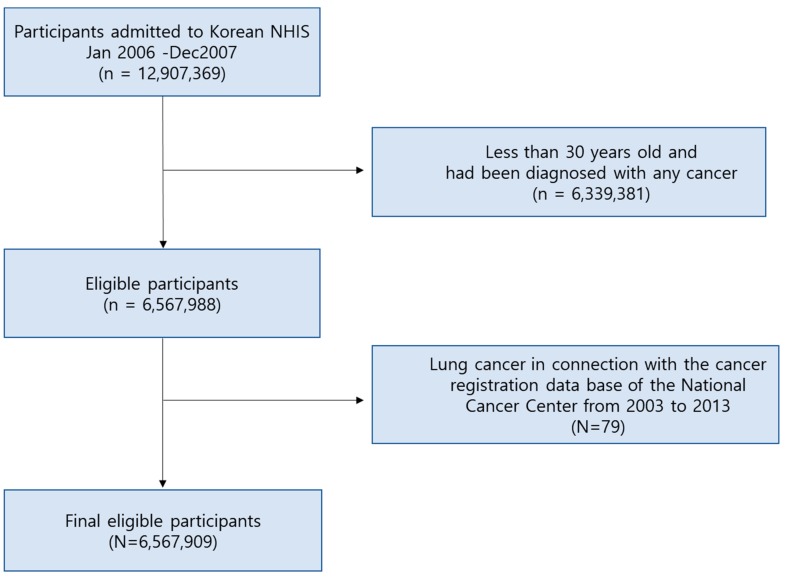
Flow chart of participants recruitment from Korean NHIS. NHIS = National Health Insurance Service.

**Table 1 ijerph-17-00915-t001:** The average exposure level of air pollutants in the past 5 years (2002–2007).

	Mean	SD	Min	25th Percentile	50th Percentile	75th Percentile	Max
PM10 (µg/m^3^)	55.80	6.30	37.99	50.40	57.30	60.90	75.95
NO_2_ (ppm)	23.90	9.00	4.16	16.30	23.50	32.10	42.66

PM10 = particulate matter less than or equal to 10 μm in a diameter, NO_2_ = nitrogen dioxide.

**Table 2 ijerph-17-00915-t002:** General characteristics of the study subjects (*n* = 6,567,909).

	*n* (%)
Age	
30–39 yr	1,860,367 (28.3)
40–49 yr	2,007,864 (30.6)
50–59 yr	1,386,946 (21.1)
60–69 yr	884,183 (13.5)
70–79 yr	369,650 (5.6)
≥80 yr	58,899 (0.9)
Gender	
Male	4,314,480 (65.7)
Female	2,253,429 (34.3)
BMI	
<18.5	168,578 (2.6)
18.5–22.9	2,402,068 (36.6)
23–24.9	1,721,017 (26.2)
25–29.9	2,063,203 (31.4)
≥30	207,735 (3.2)
Smoking	
Non-smoker	3637,984 (59.4)
Former-smoker	653,621 (10.7)
Current-smoker	1,829,330 (29.9)
Pack-year/current-smoker	17.1(10.9)
Physical activity	
None	3,168,619 (51.6)
1–2/week	1,798,894 (29.3)
3–4/week	687,586 (11.2)
5–6/week	163,074 (2.7)
Everyday	318,890 (5.2)
History of COPD	
No	6,484,490 (98.7)
Yes	83,419 (1.3)
Lung disease due to external factors	
No	6,564,745 (100.0)
Yes	3,164 (0.05)
Interstitial lung disease	
No	6,560,779 (99.9)
Yes	7130 (0.1)

Values are presented as number of patient (%).

**Table 3 ijerph-17-00915-t003:** Lung cancer incidence by histologic classification of study subjects.

Cancer Type	All (*n* = 6,567,909)		Male (*n* = 4,314,480)		Female (*n* = 2,253,429)	
	Cases	Incidence Rate(Per 100,000 Person-Years)	Cases	Incidence Rate(Per 100,000 Person-Years)	Cases	Incidence Rate(Per 100,000 Person-Years)
All lung cancer	27,518	419.0	21,912	333.6	5606	85.4
Non-small cell carcinoma	24,031	365.9	19,076	290.4	4955	75.4
Squamous cell carcinoma	7280	110.8	6912	105.2	368	5.6
Adenocarcinoma	8223	125.2	5744	87.5	2479	37.7
Large cell carcinoma	314	4.8	291	4.4	23	0.4
Other	8214	125.1	6129	93.3	2085	31.7
Small cell carcinoma	3127	47.6	2836	43.2	291	4.4

**Table 4 ijerph-17-00915-t004:** Hazard ratios of total lung cancer incidence according to the general information of the study subjects.

	Overall Lung Cancer					
	All	Case/At RiskHR (95% CI)	Male	Case/At RiskHR (95% CI)	Female	Case/At RiskHR (95% CI)
**Gender**						
Male	21,912/4,314,480	2.25(2.18–2.32)				
Female	5246/2,253,429	1.00 (ref.)				
**BMI**						
<18.5	1368/168,578	1.68(1.59–1.78)	1128/85,386	1.95(1.84–2.08)	240/83,192	1.43(1.25–1.64)
18.5–22.9	11,596/2,402,068	1.00 (ref.)	9608/1,416,418	1.00 (ref.)	1988/985,650	1.00 (ref.)
23–24.9	6722/1,721,017	0.81(0.79–0.83)	5485/1,192,940	0.68(0.65–0.70)	1237/528,077	1.16(1.08–1.24)
25–29.9	6972/2,063,203	0.70(0.68–0.72)	5392/1,483,685	0.54(0.52–0.55)	1580/579,518	1.34(1.26–1.43)
≥30	481/207,735	0.48(0.44–0.53)	285/133,202	0.32(0.28–0.36)	196/74,533	1.29(1.12–1.50)
**Smoking**						
Non-smoker	11,266/3,637,984	1.00 (ref.)	6838/1,647,730	1.00 (ref.)	4428/1,990,254	1.00 (ref.)
Former-smoker	2960/653,621	1.49(1.43–1.55)	2879/631,592	1.10(1.06–1.15)	81/22,029	1.67(1.34–2.08)
Current-smoker	11,242/1,829,330	2.04(1.99–2.10)	10,879/1,773,699	1.50(1.46–1.55)	363/55,631	2.95(2.65–3.28)
**Physical activity**						
None	15,139/3,168,619	1.00 (ref.)	11,928/1,870,962	1.00 (ref.)	3211/1,297,657	1.00 (ref.)
1–2/week	5111/1,798,894	0.60(0.58–0.62)	4367/1,384,309	0.50(0.48–0.52)	744/414,585	0.73(0.67–0.79)
3–4/week	2154/687,586	0.66(0.63–0.69)	1740/487,860	0.56(0.53–0.59)	414/199,726	0.84(0.76–0.93)
5–6/week	643/163,074	0.83(0.77–0.90)	510/107,404	0.75(0.68–0.81)	133/55,670	0.97(0.82–1.16)
Everyday	2222/318,890	1.44(1.38–1.51)	1855/201,326	1.43(1.36–1.50)	367/117,564	1.25(1.12–1.39)
**Alcohol consumption**						
None	13,152/2,956,359	1.00 (ref.)	8925/1,336,835	1.00 (ref.)	4227/1,619,524	1.00 (ref.)
2–3/month	2823/1,161,376	0.56(0.54–0.58)	2444/877,845	0.42(0.40–0.44)	379/283,531	0.52(0.47–0.58)
1–2/week	4310/1,373,243	0.72(0.70–0.75)	4092/1,220,541	0.51(0.49–0.53)	218/152,702	0.56(0.49–0.64)
3–4/week	2545/473,454	1.23(1.18–1.29)	2503/446,658	0.84(0.81–0.88)	42/26,796	0.61(0.45–0.83)
Everyday	2736/213,508	2.89(2.77–3.01)	2697/200,618	1.99(1.90–2.07)	39/12,890	1.16(0.85–1.59)
**History of COPD**						
No	24,982/6,484,490	1.00 (ref.)	19,968/4,260,123	1.00 (ref.)	5014/2,224,367	1.00 (ref.)
Yes	2176/83,419	6.64(6.35–6.94)	1944/54,357	7.47(7.13–7.82)	232/29,062	3.47(3.05–3.96)
**Lung disease due to external factors**						
No	27,058/6,564,745	1.00 (ref.)	21,815/4,311,633	1.00 (ref.)	5243/2,253,112	1.00 (ref.)
Yes	100/3164	7.60(6.24–9.24)	97/2847	6.60(5.40–8.05)	3/317	3.93(1.27–12.20)
**Interstitial lung disease**						
No	26,918/6,560,779	1.00 (ref.)	21,685/4,309,639	1.00 (ref.)	5233/2,251,140	1.00 (ref.)
Yes	240/7130	8.19(7.21–9.30)	227/4841	9.33(8.19–10.64)	13/2289	2.43(1.41–4.18)

HR = hazard ratio, CI = confidence interval.

**Table 5 ijerph-17-00915-t005:** Risk of lung cancer incidence by histological classification according to exposure to PM_10_ or NO_2_ by gender and smoking status.

	Events	PM_10_		NO_2_	
		Model1HR (95% CI)	Model2HR (95% CI)	Model1HR (95% CI)	Model2HR (95% CI)
Male					
Non-smoker					
All lung cancer	6838	0.92(0.86–0.97)	0.95(0.90–1.01)	0.85(0.80–0.90)	0.90(0.85–0.96)
Non-small cell carcinoma	6124	0.92(0.87–0.98)	0.96(0.90–1.02)	0.85(0.80–0.91)	0.90(0.85–0.96)
Squamous cell carcinoma	1900	0.87(0.78–0.97)	0.94(0.83–1.05)	0.70(0.62–0.79)	0.77(0.68–0.88)
Adenocarcinoma	2052	0.92(0.83–1.02)	0.93(0.84–1.04)	0.95(0.85–1.05)	0.98(0.88–1.09)
Large cell carcinoma	59	1.11(0.61–2.02)	1.25(0.68–2.30)	0.66(0.33–1.35)	0.68(0.32–1.44)
Other	2113	0.97(0.87–1.08)	1.00(0.90–1.11)	0.91(0.82–1.01)	0.95(0.85–1.06)
Small cell carcinoma	714	0.87(0.72–1.04)	0.90(0.75–1.09)	0.82(0.68–0.99)	0.89(0.74–1.09)
Former-smoker					
All lung cancer	2879	0.91(0.83–0.99)	0.92(0.84–1.01)	0.99(0.90–1.08)	1.03(0.94–1.13)
Non-small cell carcinoma	2569	0.93(0.84–1.02)	0.94(0.85–1.04)	0.98(0.89–1.08)	1.03(0.94–1.14)
Squamous cell carcinoma	834	0.82(0.68–0.97)	0.83(0.69–0.99)	0.84(0.71–1.00)	0.92(0.77–1.10)
Adenocarcinoma	776	1.05(0.89–1.24)	1.02(0.85–1.21)	1.20(1.02–1.40)	1.17(0.96–1.38)
Large cell carcinoma	44	0.79(0.37–1.70)	0.80(0.35–1.83)	0.54(0.23–1.27)	0.53(0.20–1.36)
Other	915	0.93(0.80–1.10)	0.99(0.84–1.17)	0.96(0.82–1.13)	1.04(0.88–1.22)
Small cell carcinoma	310	0.76(0.57–1.02)	0.76(0.56–1.03)	1.03(0.79–1.34)	1.06(0.80–1.40)
Current-smoker					
All lung cancer	10,879	1.00(0.95–1.04)	1.03(0.98–1.08)	0.97(0.92–1.02)	0.99(0.94–1.04)
Non-small cell carcinoma	9241	1.01(0.96–1.06)	1.05(0.99–1.11)	0.96(0.91–1.01)	0.98(0.93–1.04)
Squamous cell carcinoma	3755	0.97(0.90–1.05)	1.01(0.93–1.10)	0.91(0.84–0.99)	0.97(0.89–1.06)
Adenocarcinoma	2570	1.11(1.01–1.21)	1.14(1.03–1.25)	1.05(0.96–1.16)	1.03(0.93–1.14)
Large cell carcinoma	167	0.79(0.53–1.18)	0.87(0.57–1.33)	0.78(0.51–1.18)	0.83(0.53–1.30)
Other	2749	0.99(0.90–1.08)	1.02(0.93–1.13)	0.94(0.85–1.03)	0.97(0.87–1.07)
Small cell carcinoma	1638	0.91(0.80–1.03)	0.91(0.80–1.03)	1.03(0.92–1.17)	1.03(0.90–1.17)
Female					
Non-smoker					
All lung cancer	4428	0.99(0.92–1.07)	0.99(0.92–1.07)	1.00(0.93–1.08)	1.01(0.93–1.08)
Non-small cell carcinoma	4231	0.99(0.92–1.07)	0.99(0.92–1.07)	1.02(0.95–1.10)	1.03(0.95–1.11)
Squamous cell carcinoma	265	0.95(0.70–1.29)	0.97(0.71–1.32)	0.91(0.66–1.26)	0.93(0.67–1.29)
Adenocarcinoma	2164	0.94(0.85–1.05)	0.93(0.84–1.04)	0.96(0.86–1.07)	0.97(0.87–1.08)
Large cell carcinoma	16	0.55(0.12–2.41)	0.59(0.13–2.48)	1.39(0.45–4.36)	1.45(0.46–4.60)
Other	1786	1.07(0.95–1.19)	1.07(0.95–1.20)	1.12(0.99–1.25)	1.11(0.99–1.25)
Small cell carcinoma	197	0.97(0.68–1.37)	1.01(0.71–1.45)	0.52(0.33–0.82)	0.55(0.35–0.88)
Former-smoker					
All lung cancer	81	1.24(0.75–2.04)	1.47(0.87–2.48)	1.06(0.62–1.80)	1.22(0.70–2.12)
Non-small cell carcinoma	73	1.42(0.85–2.37)	1.68(0.98–2.87)	1.03(0.59–1.81)	1.15(0.65–2.07)
Squamous cell carcinoma	14	0.57(0.13–2.58)	0.67(0.14–3.13)	0.27(0.04–2.07)	0.31(0.04–2.44)
Adenocarcinoma	24	1.36(0.56–3.30)	1.41(0.56–3.54)	1.71(0.72–4.05)	1.81(0.74–4.43)
Large cell carcinoma	1	-	-	-	-
Other	34	2.01(0.99–4.08)	2.77(1.29–5.96)	1.02(0.44–2.35)	1.22(0.51–2.93)
Small cell carcinoma	8	-	-	1.35(0.27–6.76)	1.54(0.29–8.25)
Current-smoker					
All lung cancer	363	0.86(0.67–1.11)	0.81(0.62–1.07)	0.92(0.70–1.21)	0.93(0.70–1.25)
Non-small cell carcinoma	293	0.86(0.64–1.14)	0.85(0.63–1.14)	0.90(0.66–1.23)	0.97(0.70–1.34)
Squamous cell carcinoma	69	0.48(0.24–0.98)	0.54(0.27–1.11)	0.57(0.27–1.20)	0.70(0.33–1.48)
Adenocarcinoma	97	0.80(0.49–1.30)	0.72(0.42–1.21)	1.17(0.72–1.89)	1.14(0.69–1.89)
Large cell carcinoma	3	2.07(0.19–23.10)	2.36(0.21–26.88)	-	-
Other	124	1.15(0.76–1.74)	1.16(0.75–1.80)	0.90(0.55–1.47)	0.98(0.59–1.65)
Small cell carcinoma	70	0.89(0.50–1.58)	0.70(0.37–1.30)	0.99(0.54–1.83)	0.79(0.40–1.57)

HR = hazard ratio, CI = confidence interval, PM_10_ = particulate matter with aerodynamic diameter less than 10 μm, NO_2_ = nitrogen dioxide; Model 1: adjusted for age and gender; Model 2: adjusted for age, gender, income, BMI, pack-year, history of COPD, interstitial lung disease, lung disease due to external factor, physical activity and alcohol consumption.

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
