# Peer review of "Air Pollution and Incidence of Lung Cancer by Histological Type in Korean Adults: A Korean National Health Insurance Service Health Examinee Cohort Study"

_ijerph, 2020, doi:10.3390/ijerph17030915_

Round 1

Reviewer 1 Report

The article entitled “Air pollution and incidence of lung cancer by histological type in Korean adults: a Korean national health insurance service health examinee cohort study” is well put together, showing that long-term air pollution exposure is associated with an increased risk of adenocarcinoma in male smokers in Korea. Although well written, the article needs extensive English editing. In addition, I have some comments as well:

In the Abstract and the Methods section the authors identify this study as longitudinal. But then in the Discussion section, page 9, second paragraph from the bottom, they say that the study design was retrospective. They repeat this statement in the last sentence before Conclusions, by saying “This large scale retrospective cohort study…” Which one is it? Please correct the statements with the appropriate study design. In the Introduction section, first sentence, the authors write “… poor prognosis and mortality”. What means “poor mortality”??? Please rephrase. In the Introduction section, second sentence, smoking is one of the main causes for what? Need to rephrase. In the Introduction section, third sentence, when and where were those studies conducted? Need to specify. In the Introduction section, second paragraph, last sentence, when and where were those studies conducted? Need to specify. In the Materials and Methods section, Study design and participants subsection, you need to explain why you excluded from your study individuals under the age of 30. In the Materials and Methods section, Statistical analysis subsection, you need to specify that you used the Asian cutoff point for BMI, and not the general cutoff point of BMI. In the Materials and Methods section, Statistical analysis subsection, are your analyses weighted? Have you included in your analyses strata, cluster and weight variables, since your study is at the national level? You need to include this information in your paper. In Table 2 it is confusing how you used the age variable in your models: categorical or continuous. If you used it as continuous please provide the median and IQR for age as an interval (Q1, Q3) and NOT as a point estimate. On page 6, first line of text, did you mean to write “lower BMI group” instead of “lower weight group”? You need to make the right correction. In the Discussion section, page 8, first paragraph, third sentence, the numbers provided are not representing incidence but frequency. You need to make the right correction. In the Discussion section, page 8, second paragraph, second sentence, what means “…developing past smokers in smokers…”? Need to rephrase. In the Discussion section, page 8, second paragraph, second sentence, what means “… and 2.04 as a strong risk factor…”? Need to rephrase. In the Discussion section, page 8, second paragraph, third sentence, what means “… the histological type of Korea…”? Need to rephrase. In the Discussion section, page 8, second paragraph, seventh sentence, what means “… squamous CLL carcinoma…”? Need to rephrase. In the Discussion section, page 9, first paragraph, first sentence, did you mean to write “… with UP TO 10 micrometers…”? Need to rephrase. In the Discussion section, page 9, first paragraph, second sentence, need to explain what those numbers in parentheses represent. In the Discussion section, page 9, second paragraph, second sentence, you wrote “The recently published findings…” Among whom? Where? When? Need to specify. In the Discussion section, page 9, second paragraph, fourth sentence, what means “… with non-small cell lung pollution…”? Need to rephrase. In the Discussion section, page 9, second paragraph, sixth sentence, you wrote “… have shown the association with other…” The association between what and what? Need to rephrase.

Reviewer 2 Report

Authors reported the risk of lung cancers according to the air pollution such as PM10 or NO2 in Korea. Using Korean National Health Insurance Service database for 2006 -2007, they extracted more than 6 million cancer-free subjects. The lung cancers were found in 27,518. Regarding air pollution, they liked according to the home addresses for 2002 to 2007.

They found higher exposure group of PM10 and NO2 (upper 25 percentile) in male and current smoker showed significant increased incidences of adenocarcinoma when compared with bottom 75 percentile population.

Other results which authors showed in this study such as features of lung cancer patients did not much differ from previous analyses.

The following issues should be modified by authors.

1) If they have any findings from literatures, the critical relationship between air pollution (PM10)/NO2 and adeno-carcinoma in lung cancer, it should be introduced in “Discussion”.

2) If they have any compositional data regarding PM10 in their assays, it would be better to show it.

3) In table 5, the categorization is not easy to understand. All lung cancer should be divided to “Non-small cell” and “Small cell”, then, “non-small cell” should divided into “Squamous”, “Adeno” and “Large cell” (probably “Other” would be initially positioned into “Non-small cell”.

4) In the 5th paragraph in “Discussion”, the differences between this study and ref[25]’s study (similar form Korea) should be described in detail. Why, in the same nation”, the findings were so differed?

5) There is “Howevere” in p9, so, please check English again in whole manuscript,
